# Maximum Geometric Quantum Entropy

**DOI:** 10.3390/e26030225

**Published:** 2024-03-01

**Authors:** Fabio Anza, James P. Crutchfield

**Affiliations:** 1Department of Mathematics Informatics and Geoscience, University of Trieste, Via Alfonso Valerio 2, 34127 Trieste, Italy; 2Complexity Sciences Center and Physics Department, University of California at Davis, One Shields Avenue, Davis, CA 95616, USA; crutchfield@ucdavis.edu

**Keywords:** quantum mechanics, geometric quantum mechanics, maximum entropy estimation, density matrix, 05.45.-a, 89.75.Kd, 89.70.+c, 05.45.Tp

## Abstract

Any given density matrix can be represented as an infinite number of ensembles of pure states. This leads to the natural question of how to uniquely select one out of the many, apparently equally-suitable, possibilities. Following Jaynes’ information-theoretic perspective, this can be framed as an inference problem. We propose the Maximum Geometric Quantum Entropy Principle to exploit the notions of Quantum Information Dimension and Geometric Quantum Entropy. These allow us to quantify the entropy of fully arbitrary ensembles and select the one that maximizes it. After formulating the principle mathematically, we give the analytical solution to the maximization problem in a number of cases and discuss the physical mechanism behind the emergence of such maximum entropy ensembles.

## 1. Introduction

### 1.1. Background

Quantum mechanics defines a system’s state |ψ〉 as an element of a Hilbert space H. These are the *pure states*. To account for uncertainties in a system’s actual state |ψ〉, one extends the definition to *density operators* ρ that act on H. These operators are linear, positive semidefinite ρ≥0, self-adjoint $\rho =\rho^{†}$, and normalized Trρ=1. ρ, then, is a pure state when it is also a projector: $\rho^{2}=\rho$.

The spectral theorem guarantees that one can always decompose a density operator as $\rho =\sum_{i}\lambda_{i}|\lambda_{i}\rangle \langle \lambda_{i}|$, where $\lambda_{i}\in \left[0,1\right]$ are its eigenvalues and $|\lambda_{i}\rangle$ its eigenvectors. *Ensemble theory* [1,2] gives the decomposition’s statistical meaning: $\lambda_{i}$ is the probability that the system is in the pure state $|\lambda_{i}\rangle$. Together, they form ρ’s eigenensemble $L\left(\rho \right):={\left\{\lambda_{j},|\lambda_{j}\rangle \right\}}_{j}$, which, putting degeneracies aside for a moment, is unique. L(ρ), however, is not the only ensemble compatible with the measurement statistics given by ρ. Indeed, there is an infinite number of different ensembles that give the same density matrix: ${\left\{p_{k},|\psi_{k}\rangle \right\}}_{k}$ such that $\sum_{k}p_{k}|\psi_{k}\rangle \langle \psi_{k}|=\sum_{j}\lambda_{j}|\lambda_{j}\rangle \langle \lambda_{j}|$. Throughout the following, E(ρ) identifies the set of all ensembles of pure states consistent with a given density matrix.

### 1.2. Motivation

Since the association ρ→E(ρ) is one-to-many, it is natural to ask whether a meaningful criterion to uniquely select an element of E(ρ) exists. This is a typical inference problem, and a principled answer is given by the maximum entropy principle (MEP) [3,4,5]. Indeed, when addressing inference given only partial knowledge, maximum entropy methods have enjoyed marked empirical success. They are broadly exploited in science and engineering.

Following this lead, the following answers the question of uniquely selecting an ensemble for a given density matrix by adapting the maximum entropy principle. We also argue in favor of this choice by studying the dynamical emergence of these ensembles in a number of cases.

The development is organized as follows. Section 2 discusses the relevant literature on this problem. It also sets up language and notation. Section 3 gives a brief summary of Geometric Quantum Mechanics: a differential-geometric language to describe the states and dynamics of quantum systems [6,7,8,9,10,11,12,13,14,15,16,17,18,19,20,21,22,23]. Then, Section 4 introduces the technically pertinent version of MEP—the Maximum Geometric Entropy Principle (MaxGEP). Section 5 discusses two mechanisms that can lead to the MaxGEP and identifies different physical situations in which the ensemble can emerge. Eventually, Section 6 summarizes what this accomplishes and draws several forward-looking conclusions.

## 2. Existing Results

The properties and characteristics of pure-state ensembles is a vast and rich research area, one whose results are useful across a large number of fields from quantum information and quantum optics to quantum thermodynamics and quantum computing, to mention only a few. This section discusses four sets of results relevant to our purposes. This also allows introducing language and notation.

First, recall Ref. [24], where Hughston, Josza, and Wootters gave a constructive characterization of all possible ensembles behind a given density matrix, assuming an ensemble with a finite number of elements. Second, Wiseman and Vaccaro, in Ref. [25], then argued for a preferred ensemble via the dynamically motivated criterion of a *Physically Realizable* ensemble. Third, Goldstein, Lebowitz, Tumulka, and Zanghi singled out the Gaussian Adjusted Projected (GAP) measure as a preferred ensemble behind a density matrix in a thermodynamic and statistical mechanics setting [26]. Fourth, Brody and Hughston used one form of maximum entropy within geometric quantum mechanics [27].

### 2.1. HJW Theorem

At the technical level, one of the most important results for our purposes is the Hughston–Josza–Wootters (HJW) theorem, proved in Ref. [24], which we now summarize.

Consider a system with finite-dimensional Hilbert space $H_{S}$ described by a density matrix ρ with rank *r*: $\rho =\sum_{j=1}^{r}\lambda_{j}|\lambda_{j}\rangle \langle \lambda_{j}|$. We assume $\dim\;H_{S}:=d_{S}=r$, since the case in which $d_{S}>r$ is easily handled by restricting $H_{S}$ to the *r*-dimensional subspace defined by the image of ρ. Then, a generic ensemble $e_{\rho }\in E\left(\rho \right)$ with $d\geq d_{S}$ elements can be generated from L(ρ) via linear remixing with a $d\times d_{S}$ matrix *M* having as columns $d_{S}$ orthonormal vectors. Then, $e_{\rho }=\left\{p_{k},|\psi_{k}\rangle \right\}$ is given by the following:$$\begin{array}{c} \sqrt{p_{k}}|\psi_{k}\rangle =\sum_{j=1}^{d_{S}}M_{kj}\sqrt{\lambda_{j}}|\lambda_{j}\rangle \;. \end{array}$$

Equivalently, one can generate ensembles applying a generic d×d unitary matrix *U* to a list of *d* non-normalized $d_{S}$-dimensional states in which the first $d_{S}$, ${\left\{\sqrt{\lambda_{j}}|\lambda_{j}\rangle \right\}}_{j=1}^{d_{S}}$, are proportional to the eigenvectors of ρ, while the remaining $d-d_{S}$ are simply null vectors:$$\begin{array}{c} \sqrt{p_{k}}|\psi_{k}\rangle =\sum_{j=1}^{d_{S}}U_{kj}\sqrt{\lambda_{j}}|\lambda_{j}\rangle \;. \end{array}$$

Here, we must remember that *U* is not an operator acting on $H_{S}$ but a unitary matrix mixing weighted eigenvectors into *d* non-normalized vectors.

The power of the HJW theorem is not only that it introduces a constructive way to build E(ρ) ensembles but that this way is complete. Namely, all ensembles can be built in this way. This is a remarkable fact, which the following sections rely heavily on.

### 2.2. Physically Realizable Ensembles

For our purposes, a particularly relevant result is that of Wiseman and Vaccaro [25]. (See also subsequent results by Wiseman and collaborators on the same topic [28]). The authors argue for a *Physically Realizable* ensemble that is implicitly selected by the fact that if a system is in a stationary state $\rho_{ss}$, one would like to have an ensemble that is stable under the action of the dynamics generated by monitoring the environment. This is clearly desirable in experiments in which one monitors an environment to infer properties about the system. While this is an interesting way to answer the same question we tackle here, their answer is based on dynamics and limited to stationary states. The approach we propose here is very different, being based on an inference principle. This opens interesting questions related to understanding the conditions under which the two approaches provide compatible answers. Work in this direction is ongoing and it will be reported elsewhere.

### 2.3. Gaussian Adjusted Projected Measure

Reference [26] asks a similar question to that here but in a statistical mechanics and thermodynamics context. Namely, viewing pure states as points on a high-dimensional sphere $\psi \in S^{2d_{S}-1}$, which probability measure μ on $S^{2d_{S}-1}$, interpreted as a *smooth* ensemble on $S^{2d_{S}-1}$, leads to a thermal density matrix:$$\begin{array}{c} \rho_{th}\overset{?}{=}\int d\psi \mu \left(\psi \right)|\psi \rangle \langle \psi | \end{array}$$

Here, $\rho_{th}$ could be the microcanonical or the canonical density matrix. Starting with Schrödinger’s [29,30] and Bloch’s [31] early work, the authors argue in favor of the Gaussian Adjusted Projected (GAP) measure. This is essentially a Gaussian measure, adjusted and projected to live on $\psi \in S^{2d_{S}-1}$:$$\begin{array}{c} GAP\left(\sigma \right)\propto e^{-\langle \psi |\sigma^{-1}|\psi \rangle }\;. \end{array}$$

Written explicitly in terms of complex coordinates $\psi_{j}$, it is clear that this is a Gaussian measure with vanishing average $E[\psi_{j}]=0$ and covariance specified by $E\left[\psi_{j}^{*}\psi_{k}\right]=\sigma_{jk}$. In particular, σ=ρ guarantees that GAP(ρ) has ρ as density matrix.

The GAP measure has some interesting properties [26,32,33] and, as we see in Section 4, it is also closely related to one of our results in a particular case. Our results can therefore be understood as a generalization of the GAP measure. We will not delve deeper into this matter now but comment on it later.

### 2.4. Geometric Approach

In 2000, Brody and Hughston performed the first maximum entropy analysis for the ensemble behind the density matrix [27], in a language and spirit that is quite close to those we use here. Their result came before the definition of the GAP measure, but it is essentially identical to it: $\mu \left(\psi \right)\propto \exp\left(-\sum_{j,k}L_{jk}\psi_{j}^{*}\psi_{k}\right)$. Their perspective, however, is very different from that in Ref. [26], which is focused on thermal equilibrium phenomenology. The work we perform here, and our results, can also be understood as a generalization of Ref. [27]. Indeed, as we argued in Ref. [34] (and will show again in Section 4), the definition of entropy used (see Equation (10) in Ref. [27]) is meaningful only in certain cases, in particular, when the ensemble has support with dimension equal to the dimension of the state space of the system of interest. In general, more care is required.

### 2.5. Summary

We summarized four relevant sets of results on selecting one ensemble among the infinitely many that are generally compatible with a density matrix. Our work relies heavily on the HJW theorem [24], and it is quite different from the approach by Wiseman and Vaccaro [25]. Moreover, it constitutes a strong generalization with respect to the results on the GAP measure [26] in a thermal equilibrium context and with respect to the analysis by Brody and Hughston in [27].

## 3. Geometric Quantum States

Our maximum geometric entropy principle relies on a differential-geometric approach to quantum mechanics called *Geometric Quantum Mechanics* (GQM). The following gives a quick summary of GQM and how its notion of Geometric Quantum State [6,7,34] can be elegantly used to study physical and information-theoretic aspects of ensembles. More complete discussions are found in the relevant literature [8,9,10,11,12,13,14,15,16,17,18,19,20,21,22,23].

### 3.1. Quantum State Space

The state space of a finite-dimensional quantum system with Hilbert space $H_{S}$ is a projective Hilbert space $P(H_{S})$, which is isomorphic to a complex projective space $P\left(H_{S}\right)∼CP^{d_{S}-1}:=\left\{Z\in C^{d_{S}}:Z∼\lambda Z,\lambda \in C/0\right\}$. Pure states are thus in one-to-one correspondence with points $Z\in CP^{d_{S}-1}$. Using a computational basis as reference basis ${\left\{|j\rangle \right\}}_{j=1}^{d_{S}}$, *Z* has homogeneous coordinates $Z=(Z^{1},\ldots ,Z^{d_{S}})$ where $|Z\rangle =\sum_{j=1}^{d_{S}}Z^{j}|j\rangle \in H_{S}$. One of the advantages in using the geometric approach is that one can exploit the symplectic character of the state space. Indeed, this implies that the quantum state space $CP^{n}$ can essentially be considered as a classical, although curved, phase space. With probability and phases being canonically conjugated coordinates, $Z^{j}=\sqrt{p_{j}}e^{i\phi_{j}}$, we have $\left\{p_{j},\phi_{k}\right\}=\delta_{jk}$. The intuition from classical mechanics can then be used to understand the phenomenology of quantum systems.

### 3.2. Observables

Within GQM, observables are Hermitian functions from $CP^{d_{S}-1}$ to the reals:$$\begin{array}{c} f_{O}\left(Z\right):=\sum_{j,k=1}^{d_{S}}{Z^{j}}^{*}Z^{k}O_{jk}/\sqrt{\sum_{h=1}^{d_{S}}{\left|Z^{h}\right|}^{2}}\;, \end{array}$$
where $O_{jk}=\langle j|O|k\rangle$ are the matrix elements of the Hilbert space self-adjoint operator O. An analogous relation holds for Positive Operator-Valued Measures (POVMs).

### 3.3. Geometric Quantum States

The quantum state space $P(H_{S})$ has a preferred metric $g_{FS}$ and a related volume element $dV_{FS}$—the *Fubini–Study volume element*. The details surrounding these go beyond our present purposes. It is sufficient to give $dV_{FS}$’s explicit form in the coordinate system we use for concrete calculations. This is the “probability + phase” coordinate system, given by $Z↔{\left\{(p_{j},\phi_{j})\right\}}_{j=1}^{d_{S}}$:$$\begin{array}{cc} dV_{FS} & =\sqrt{\det g_{FS}}\prod_{j=1}^{d_{S}}dZ^{j}{dZ^{j}}^{*}=\prod_{j=1}^{d_{S}}\frac{dp_{j}d\phi_{j}}{2}\;. \end{array}$$

This volume element can be used to define integration. Indeed, calling Vol[B] the volume of a set $B\subseteq P(H_{S})$, we have $\mathrm{Vol}\left[B\right]:=\int_{B}dV_{FS}$. In turn, this provides the fundamental, unitarily invariant, notion of a uniform measure on the quantum state space. This is the normalized Haar measure $\mu_{\mathrm{Haar}}$:$$\begin{array}{c} \mu_{\mathrm{Haar}}\left[B\right]=\mathrm{Vol}\left[B\right]/\mathrm{Vol}\left[P\left(H_{S}\right)\right]\;. \end{array}$$

$\mu_{\mathrm{Haar}}$ is a probability measure that weights all pure states uniformly with the total Fubini–Study volume of the quantum state space. Probability measures [35,36] are the appropriate mathematical formalization behind the physical notion of ensembles and they formalize the concept of a Geometric Quantum State (GQS): A probability measure on the (complex projective) quantum state space. For example, a pure state corresponds to a Dirac measure $\delta_{\psi }$ with support on a single point $\psi \in P(H_{S})$, with Hilbert space representation |ψ〉.

### 3.4. GQS as Conditional Probability Measures

One way to embed the HJW theorem in this geometric context is the following.

Any density matrix can be purified in an infinite number of different ways. A purification |ψ(ρ)〉 of ρ is a pure state in a larger Hilbert space $|\psi \left(\rho \right)\rangle \in H_{S}⊗H_{E}$ such that ${\mathrm{Tr}}_{E}\left|\psi \left(\rho \right)\rangle \langle \psi \left(\rho \right)\right|$, where ${\mathrm{Tr}}_{E}$ is the partial trace over the additional Hilbert space $H_{E}$. It is known that, for the purification to be achieved, $d_{E}\geq r$. Since we assume $r=d_{S}$, we have $d_{E}\geq d_{S}$. Any purification of ρ will have a Schmidt decomposition of a specific type:$$\begin{array}{c} |\psi \left(\rho \right)\rangle =\sum_{j=1}^{d_{S}}\sqrt{\lambda_{j}}|\lambda_{j}\rangle |SP_{j}\rangle \;, \end{array}$$
where ${\left\{|SP_{j}\rangle \in H_{E}\right\}}_{j=1}^{d_{S}}$ are the $d_{S}$ orthonormal “Schmidt partners”. These can be extended to a full orthonormal basis on $H_{E}$ by adding $d_{E}-d_{S}$ orthonormal vectors which are orthogonal to $\mathrm{span}({\left\{|SP_{j}\rangle \right\}}_{j})$.

L(ρ) is therefore understood as the ensemble resulting from conditioning on the Schmidt partners. Namely, when measuring the environment in the basis ${\left\{|SP_{j}\rangle \right\}}_{j=1}^{d_{E}}$, the state of the system after the measurement will be $|\lambda_{j}\rangle$ with probability $\lambda_{j}$. Its GQS is $\mu_{L}=\sum_{j=1}^{d_{S}}\lambda_{j}\delta_{\lambda_{j}}$. If we now measure the environment in a generic basis, instead of using the Schmidt partners, we generate a different ensemble. Calling ${\left\{|v_{\alpha }\rangle \right\}}_{\alpha =1}^{d_{E}}$ one such basis, we have:$$\begin{array}{c} \sum_{j=1}^{d_{S}}\sqrt{\lambda_{j}}|\lambda_{j}\rangle |SP_{j}\rangle =\sum_{\alpha =1}^{d_{E}}\sqrt{p_{\alpha }}|\chi_{\alpha }\rangle |v_{\alpha }\rangle \;, \end{array}$$
with $\sqrt{p_{\alpha }}|\chi_{\alpha }\rangle =I_{S}⊗|v_{\alpha }\rangle \langle v_{\alpha }|\psi \left(\rho \right)\rangle$, ${\left\{p_{\alpha },|\chi_{\alpha }\rangle \right\}}_{\alpha =1}^{d_{E}}\in E\left(\rho \right)$, and GQS $\mu =\sum_{\alpha =1}^{d_{E}}p_{\alpha }\delta_{\chi_{\alpha }}$.

Starting from the Schmidt partners, these bases are in one-to-one correspondence with $d_{E}\times d_{E}$ unitary matrices acting on $H_{E}$: $|v_{\alpha }\rangle :=U|SP_{\alpha }\rangle$. And these, in turn, are in one-to-one correspondence with the unitary matrices in the HJW theorem. Therefore, they are an analogously complete classification of ensembles. The reason for this slight rearrangement of things with respect to the HJW theorem is that we now have an interpretation of $|\chi_{\alpha }\rangle$ as the conditionally pure state of the system, conditioned on the fact that we make a projective measurement ${\left\{|v_{\alpha }\rangle \right\}}_{\alpha =1}^{d_{E}}$ on the environment where the result α occurs with probability $p_{\alpha }$.

### 3.5. Quantifying Quantum Entropy

To develop entropy, the following uses the setup in Refs. [6,7,34] to study the physics of ensembles using geometric quantum mechanics. Since the focus here is a maximum entropy approach to select an ensemble behind a density matrix, it is important to have a proper understanding of how to quantify the entropy of an ensemble or, equivalently, of the GQS.

First, we look at the statistical interpretation of the pure states which participate in the conditional ensembles $\left\{p_{\alpha },|\chi_{\alpha }\rangle \right\}$. The corresponding kets $\left\{|\chi_{\alpha }\rangle \right\}$ are not necessarily orthogonal $\langle \chi_{\alpha }|\chi_{\beta }\rangle \neq \delta_{\alpha \beta }$, so the states are not mutually exclusive or distinguishable in the Fubini–Study sense. However, these states come with the classical labels $\alpha \to |\chi_{\alpha }\rangle$ associated with the outcomes of projective measurements on the environment. In this sense, if α≠β, we have a classical way to distinguish them, and thus we can understand how to interpret expressions like $-\sum_{\alpha }p_{\alpha }\log p_{\alpha }$.

Then, we highlight that the correct functional to use to evaluate the entropy of μ is not always the same. It depends on another feature of the ensemble, the *quantum information dimension*, which is conceptually related to the dimension of its support in quantum state space. To illustrate the concept, consider the following four GQSs of a qubit:$$\begin{array}{cccc} \; & \mu_{1}=\delta_{\psi } & \; & \mu_{2}=\sum_{k}p_{k}\delta_{\psi_{k}}\\ \; & \mu_{3}=\frac{1}{T}\int_{0}^{T}\;\;dt\delta_{\psi (t)} & \; & \mu_{4}=\mu_{\mathrm{Haar}}\;. \end{array}$$

Naturally, the entropy of $\mu_{1}$ vanishes, since there is no uncertainty. The system inhabits only one pure state, ψ. The entropy of $\mu_{2}$ is already nontrivial to evaluate. Indeed, while one obvious way is to use the functional $-\sum_{k}p_{k}\log p_{k}$, it is also very clear that this notion of entropy does not take into account the location of the points $\psi_{k}\in P\left(H_{S}\right)$. Intuitively, if all these points are close to each other, we would like our entropy to be smaller than in the case in which all the points are uniformly distributed on $\psi_{k}\in P\left(H_{S}\right)$.

The entropy of $\mu_{3}$ is perhaps the most peculiar, but it illustrates the points in the best way. Let us assume that our qubit is evolving with a Hamiltonian *H* such that $E_{1}-E_{0}=\hbar \omega$. Then, $\psi \left(t\right)=\left(\sqrt{1-p_{0}},\sqrt{p_{0}}e^{i\phi_{0}-\omega t}\right)$. If we aggregate the time average and look at the statistics we obtain, it is clear that the variable *p* is a conserved quantity—$p\left(t\right)=p_{0}$, while $\phi \left(t\right)=\phi_{0}-\omega t$ is an angular variable that, over a long time, will be uniformly distributed in [0,2π]. This means $\lim_{T\to \infty }\mu_{3}=\frac{1}{2\pi }\delta_{p_{0}}^{p}$, where $\delta_{p_{0}}^{p}$ is a Dirac measure over the first variable *p* with support on $p=p_{0}$. How do we evaluate the entropy of $\mu_{3}$?

While to evaluate $\mu_{4}=\mu_{\mathrm{Haar}}$ we simply integrate over the whole state space and obtain $\log\mathrm{Vol}(CP^{1})$, this does not work for $\mu_{3}$. Indeed, with respect to the full, 2D quantum state space (p,ϕ)∈[0,1]×[0,2π], the distribution clearly lives on a 1D line, which is a measure-zero subset.

To properly address all these different cases, a more general approach is needed. Reference [34] adapted previous work by Renyi to probability measures on a quantum state space. This led to the notions of *Quantum Information Dimension* D and *Geometric Quantum Entropy* $H_{D}$ that address these issues and properly evaluate the entropy in all these cases. We now give a quick summary of the results in Ref. [34].

### 3.6. Quantum Information Dimension and Geometric Entropy

Thanks to the symplectic nature of $P(H_{S})$, the quantum state space is essentially a curved, compact, classical phase space. We can therefore apply classical statistical mechanics to it, using ${\left\{(p_{j},\phi_{j})\right\}}_{j=1}^{d_{S}}$ as canonical coordinates. Since the Fubini–Study volume is $dV_{FS}\propto \prod_{j}dp_{j}d\phi_{j}$, we can coarse-grain $P(H_{S})$ by partitioning it into phase-space cells $C_{\vec{a},\vec{b}}$:$$\begin{array}{c} C_{\vec{a}\vec{b}}=\prod_{j=1}^{d_{S}-1}\left[\frac{a_{j}}{N},\frac{a_{j}+1}{N}\right]\times 2\pi \left[\frac{b_{j}}{N},\frac{b_{j}+1}{N}\right] \end{array}$$
of equal Fubini–Study volume $\mathrm{Vol}\left[C_{\vec{a}\vec{b}}\right]=\frac{\mathrm{Vol}[P\left(H_{S}\right)]}{N^{2(d_{S}-1)}}=\epsilon^{-2(d_{S}-1)}$, where $\vec{a}=\left(a_{1},\ldots ,a_{d_{S}-1}\right),$ $\vec{b}=\left(b_{1},\ldots ,b_{d_{S}-1}\right)$ and $a_{j},b_{j}=0,1,\ldots ,N$.

The coarse-graining procedure produces a discrete probability distribution $q_{\vec{a}\vec{b}}:=\mu \left[C_{\vec{a}\vec{b}}\right]$, for which we can compute the Shannon entropy:$$\begin{array}{c} H\left[\epsilon \right]:=-\sum_{\vec{a},\vec{b}}q_{\vec{a}\vec{b}}\log q_{\vec{a}\vec{b}}\;. \end{array}$$

As we change ϵ=1/N→0, the degree of coarse-graining changes accordingly. The scaling behavior of H[ϵ] provides structural information about the underlying ensemble. Indeed, since one can prove that for ϵ→0, H[ϵ] has asymptotics
$$\begin{array}{c} H\left[\epsilon \right]{∼}_{\epsilon \to 0}D\left(-\log\epsilon \right)+h_{D}\;, \end{array}$$
two quantities define its scaling behavior: D is the quantum information dimension and $h_{D}$ is the geometric quantum entropy. Their explicit definitions are:
(1a)$$\begin{array}{cc} \; & D:=\lim_{\epsilon \to 0}\frac{H[\epsilon ]}{-\log\epsilon }\;, \end{array}$$
(1b)$$\begin{array}{cc} \; & h_{D}:=\lim_{\epsilon \to 0}\left(H[\epsilon ]+D\log\epsilon \right)\;. \end{array}$$

Note how this keeps the dependence of the entropy on the information dimension explicit. This clarifies how, only in certain cases, one can use the continuous counterpart of Shannon’s discrete entropy. In general, its exact form depends on the value of D and it cannot be written as an integral on the full quantum state space with the Fubini–Study volume form.

## 4. Principle of Maximum Geometric Quantum Entropy

This section presents a fine-grained characterization of selecting an ensemble behind a given density matrix. This leverages both the HJW theorem and previous results by the authors. First, we note that D foliates E(ρ) into non-overlapping subsets $E_{D}\left(\rho \right)$ collecting all ensembles μ at given density matrix ρ and with information dimension D:$$\begin{array}{cccc} \; & E_{D}\left(\rho \right)\cap E_{D^{\prime }}\left(\rho \right)=\delta_{D,D^{\prime }}E_{D}\left(\rho \right)\;, & \; & E\left(\rho \right)=\cup_{D}E_{D}\left(\rho \right)\;. \end{array}$$

As argued above, ensembles with different D pertain to different physical situations. These can be wildly different. Therefore, we often want to first select the D of the ensemble we will end up with and then choose that with the maximum geometric entropy. Thus, here we introduce the principle of maximum geometric entropy at fixed information dimension.

**Proposition** **1**(Maximum Geometric Entropy Principle). *Given a system with density matrix ρ, the ensemble $\mu_{ME}^{D}$ that makes the fewest assumptions possible about our knowledge of the ensemble among all elements of E(ρ) with fixed information dimension dimension D is given by:*
$$\begin{array}{c} \mu_{ME}^{D}:=\underset{\mu \in E_{D}\left(\rho \right)}{arg max}h_{D}\;. \end{array}$$

Several general comments are in order. First, we note that $\mu_{ME}^{D}$ might not be unique. This should not come as a surprise. For example, with degeneracies, even the eigenensemble is not unique. Second, the optimization problem defined above is clearly constrained: the resulting ensemble has to be normalized and the average of ${\left(Z^{j}\right)}^{*}Z^{k}$ must be $\rho_{jk}$. Calling $E_{\mu }\left[A\right]$ the state space average of a function *A* performed with the GQS μ, these two constraints can be written as $C_{1}:=E_{\mu }\left[1\right]-1=0$ and $C_{jk}^{\rho }:=E_{\mu }\left[{\left(Z^{j}\right)}^{*}Z^{k}\right]-\rho_{jk}=0$. Using Lagrange multipliers, we optimize $\Lambda [\mu ,\gamma_{1},\left\{\gamma_{jk}\right\}]$ defined as:$$\begin{array}{c} \Lambda \left[\mu ,\gamma_{1},\left\{\gamma_{jk}\right\}\right]:=h_{D}\left[\mu \right]+\gamma_{1}C_{1}+\sum_{j,k}\gamma_{jk}C_{jk}^{\rho }\;. \end{array}$$

While the vanishing of Λ’s derivatives with respect to the Lagrange multipliers $\gamma_{1},\gamma_{jk}$ enforces the constraints $C_{1}=C_{jk}^{\rho }=0$, derivatives with respect to μ give the equation whose solution is the desired ensemble $\mu_{ME}^{D}$. We also note that the $\left\{\gamma_{jk}\right\}$ are not all independent. This is due to the fact that ρ is not an arbitrary matrix: Trρ=1, ρ≥0, and $\rho^{†}=\rho$. A similar relation holds for $\gamma_{jk}$.

To illustrate its use, we now solve this optimization problem in a number of relevant cases. In discussing them, it is worth introducing additional notation. Since we often use canonically conjugated coordinates, ${\left\{(p_{j},\phi_{j})\right\}}_{j=1}^{d_{S}}$, we introduce vector notation $\left(\vec{p},\vec{\phi }\right)$, with $\vec{p}\in \Delta_{d_{S}-1}$ and $\vec{\phi }\in T^{d_{S}-1}$, where $\Delta_{d_{S}-1}$ is the $\left(d_{S}-1\right)$-dimensional probability simplex and $T_{d_{S}-1}$ is the $\left(d_{S}-1\right)$-dimensional torus. Analogously, we introduce the Dirac measures $\delta_{\vec{x}}^{\vec{p}}$ and $\delta_{\vec{\varphi }}^{\vec{\phi }}$ with support on $\vec{x}\in \Delta_{d_{S}}$ and $\vec{\varphi }\in T_{d_{S}}$, respectively.

### 4.1. Finite Environments: D=0

If D=0, then the support of the ensemble is made by a number of points, which is a natural number. That is, there exists N∈N such that $\mu_{ME}^{D=0}=\sum_{\alpha =1}^{N}p_{\alpha }\delta_{\chi_{\alpha }}$, with $h_{0}=-\sum_{\alpha =1}^{N}p_{\alpha }\log p_{\alpha }$. Note how this is the HJW theorem’s domain of applicability, and this allows us to give a constructive solution.

We start by noting that *N* also foliates $E_{D=0}$ into non-overlapping sets in which the ensemble consists of *exactly N* elements. We call this set $E_{0,N}\left(\rho \right)$ and it is such that $E_{0,N}\left(\rho \right)\cap E_{0,N^{\prime }}\left(\rho \right)=\delta_{N,N^{\prime }}E_{0,N}\left(\rho \right)$, with $E_{D=0}\left(\rho \right)=\cup_{N\geq d_{S}}E_{0,N}\left(\rho \right)$. Within $E_{0,N}\left(\rho \right)$, we can use the HJW theorem with the interpretation in which the ensemble is the conditional ensemble. Here, $p_{\alpha }$ and $\chi_{\alpha }$ are generated by creating a purification of dimension *N*, in which the first $d_{S}$ elements of the basis ${\left\{|SP_{j}\rangle \right\}}_{j=1}^{d_{S}}$ are fixed and the remaining $N-d_{S}$ are free. We denote the entire basis of this type with the same symbol but a different label: ${\left\{|SP_{\alpha }\rangle \right\}}_{\alpha =1}^{N}$. The ensemble we obtain if we measure it is the eigenensemble L(ρ).

However, measuring in a different basis yields a general ensemble, with probabilities $p_{\alpha }=\langle \psi \left(\rho \right)|I_{S}⊗|v_{\alpha }\rangle \langle v_{\alpha }|\psi \left(\rho \right)\rangle =\sum_{j=1}^{d_{S}}\lambda_{j}{\left|\langle SP_{j}|v_{\alpha }\rangle \right|}^{2}$ and states $|\chi_{\alpha }\rangle =\sum_{j=1}^{d_{S}}\sqrt{\lambda_{j}}\frac{\langle v_{\alpha }|SP_{j}\rangle }{\sqrt{p_{\alpha }}}|\lambda_{j}\rangle$. With $h_{0}=-\sum_{\alpha }p_{\alpha }\log p_{\alpha }$, the absolute maximum is attained at $p_{\alpha }=1/N$. We now show, constructively, that this is always achievable while still satisfying the constraints $C_{1}=C_{ij}^{\rho }=0$, thus solving the maximization problem.

This is achieved by measuring the environment in a basis that is unbiased with respect to the Schmidt partner basis:(2)$$|v_{\alpha }\rangle :\langle v_{\alpha }|SP_{\beta }\rangle =\frac{e^{i\theta_{\alpha \beta }}}{\sqrt{N}}\;\forall \alpha ,\beta =1,\ldots ,N.$$

One such basis can always be built starting from ${\left\{|SP_{\alpha }\rangle \right\}}_{\alpha }$ by exploiting the properties of the Weyl–Heisenberg matrices via the clock-and-shift construction [37]. This is true for all N∈N. When $N=\prod_{k}n_{k}^{N_{k}}$ with $n_{k}$ primes and $N_{k}$ some integers, the finite-field algorithm [38,39] can be used to build a whole suite of *N* bases that are unbiased with respect to the Schmidt partner basis. This leads to $|\chi_{\alpha }\rangle =\sum_{j=1}^{d_{S}}\sqrt{\lambda_{j}}e^{i\theta_{\alpha j}}|\lambda_{j}\rangle$ and to:$$\begin{array}{cccc} \; & \mu_{ME}^{D=0}=\delta_{\vec{\lambda }}^{\vec{p}}\frac{1}{N}\sum_{\alpha =1}^{N}\delta_{{\vec{\theta }}_{\alpha }}^{\vec{\phi }}\;, & \; & h_{0}=\log N\;, \end{array}$$
with $\vec{\lambda }=\left(\lambda_{1},\ldots ,\lambda_{d_{S}}\right)$ and ${\vec{\theta }}_{\alpha }=\left(\theta_{\alpha 0},\ldots ,\theta_{\alpha d_{S}}\right)$.

To conclude this subsection, we simply have to show that this ensemble satisfies the constraints: $C_{1}=0$ and that the density matrix given by $\mu_{ME}^{D=0}$ is ρ, giving $C_{jk}^{\rho }=0$:$$\begin{array}{cc} {\left[\sigma_{ME}\right]}_{jk} & :=E_{\mu_{ME}^{D=0}}\left[{\left(Z^{j}\right)}^{*}Z^{k}\right]=\frac{\sqrt{\lambda_{j}\lambda_{k}}}{N}\sum_{\alpha =1}^{N}e^{i(\theta_{\alpha j}-\theta_{\alpha k})}\\ \;\; & \;=\sqrt{\lambda_{j}\lambda_{k}}\delta_{jk}=\lambda_{k}\delta_{jk}=\rho_{jk}\;. \end{array}$$

Here, the key property used is that $\frac{1}{N}\sum_{\alpha =1}^{N}e^{i(\theta_{\alpha \gamma }-\theta_{\alpha \beta })}=\langle SP_{\beta }|\sum_{\alpha =1}^{N}|v_{\alpha }\rangle \langle v_{\alpha }||SP_{\gamma }\rangle =\delta_{\beta \gamma }$, which comes from Equation (Equation 2) and the fact that ${\left\{|v_{\alpha }\rangle \right\}}_{\alpha }$ is a basis.

### 4.2. Full Support: $D=2(d_{S}-1)$

The second case of interest is the one in which the quantum information dimension takes the maximum value possible, namely $D=2(d_{S}-1)$. Then, the GQS’s support has the same dimension as the full quantum state space and the optimization problem is also tractable. This is indeed the case solved by Brody and Hughston [27]. We do not reproduce the treatment here, which is almost identical in the language of GQM. Rather, we discuss some of its physical aspects from the perspective of conditional ensembles.

If $D=2(d_{S}-1)$ and there are no other constraints aside from $C_{1}$ and $C_{jk}$, the measure $\mu_{ME}^{2(d_{S}-1)}$ can be expressed as an integral with a density $q_{ME}$ with respect to the uniform, normalized, Fubini–Study measure $dV_{FS}$:$$\begin{array}{c} \mu_{ME}^{2(d_{S}-1)}\left[A\right]=\int_{A}\;\;dV_{FS}\;q_{ME}\left(Z\right)\;. \end{array}$$

And its geometric entropy $h_{2(d_{S}-1)}$ is the continuous counterpart of Shannon’s functional on the quantum state space:$$\begin{array}{c} h_{2(d_{S}-1)}=-\int_{P(H_{S})}\;\;\;\;\;\;dV_{FS}\;q\left(Z\right)\log q\left(Z\right)\;. \end{array}$$

This was proven in Ref. [34]. Hereafter, with a slight abuse of language, we refer to both $\mu_{ME}^{2(d_{S}-1)}$ and the density $q_{ME}\left(Z\right)$ as an ensemble or the GQS.

The maximization problem leads to: $$\begin{array}{cc} \; & q_{ME}\left(Z\right)=\frac{1}{Q_{2(d_{S}-1)}\left(\rho \right)}e^{-\sum_{jk}\gamma_{jk}{\left(Z^{j}\right)}^{*}Z^{k}}\;,\\ \; & Q_{2(d_{S}-1)}\left(\rho \right)\int_{P(H_{S})}dV_{FS}\;\;e^{-\sum_{jk}\gamma_{jk}{\left(Z^{j}\right)}^{*}Z^{k}} \end{array}$$
and Lagrange multipliers $\left\{\gamma_{jk}\right\}$ are the solution of the nonlinear equations $-\frac{\partial \;\log Q}{\partial \gamma_{jk}}=\rho_{jk}$. We note how using as reference basis the eigenbasis ${\left\{|\lambda_{j}\rangle \right\}}_{j=1}^{d_{S}}$ of ρ and $Z↔(\vec{p},\vec{\phi })$ as coordinate system reveals that $\frac{\partial \;\log Q}{\partial \gamma_{jk}}=0$ when j≠k and $-\frac{\partial \;\log Q}{\partial \gamma_{jj}}=\lambda_{j}$. Thus, in this coordinate system the dependence of $\mu_{ME}^{2(d_{S}-1)}$ on the off-diagonal Lagrange multipliers disappears and we retain only the diagonal ones $\gamma_{jj}$.

Moving to a single label $\gamma_{jj}\to \gamma_{j}$ and using a vector notation:$$\begin{array}{cc} \; & q_{ME}^{2(d_{S}-1)}\left(\vec{p},\vec{\phi }\right)=\frac{1}{Q_{2(d_{S}-1)}\left(\vec{\tau }\right)}e^{\vec{\tau }\cdot \vec{p}}\;,\\ \; & Q_{2(d_{S}-1)}\left(\vec{\tau }\right)\int_{\Delta_{d_{S}-1}}\;\;\;\;\;\;d\vec{p}\;\;e^{\vec{\tau }\cdot \vec{p}}\\ \; & \vec{\tau }\left(\gamma_{d_{S}}-\gamma_{1},\gamma_{d_{S}}-\gamma_{2},\ldots ,\gamma_{d_{S}}-\gamma_{d_{S}-1}\right) \end{array}$$

Here, $Q(\vec{\tau })$ is the normalization function (a partition function). Its exact expression can be derived analytically and it is given in Appendix A.

We can see how $\mu_{ME}^{2(d_{S}-1)}$ is the product of an exponential measure on the probability simplex $\Delta_{d_{S}-1}$ and the uniform measure on the high-dimensional torus of the phases $T_{d_{S}-1}$. This leads to the following geometric entropy $h_{2(d_{S}-1)}$:(3)$$\begin{array}{cc} \; & h_{2(d_{S}-1)}\left(\vec{\tau }\right)=\log Q_{2(d_{S}-1)}\left(\vec{\tau }\right)-\vec{\tau }\cdot \vec{\lambda }\;. \end{array}$$

In this case the explicit expression of the Lagrange multipliers $\vec{\tau }$ satisfying the constraints, which was previously unknown, can be found analytically. This is reported in Appendix B.

We note that this exponential distribution on the probability simplex was recently proposed within the context of statistics and data analysis in Ref. [40]. Moreover, the exponential form associated to the maximum entropy principle is reminiscent of thermal behavior. Indeed, the shape of this distribution is closely related to the geometric canonical ensemble; see Refs. [7,14,27]. However, the value of the Lagrange multipliers is set by a different constraint, in which we fix the average energy rather than the whole density matrix.

### 4.3. Integer, but Otherwise Arbitrary, D

While one expects D to be an integer, there are GQSs that have fractal support, thus exhibiting a noninteger D. This was shown in Ref. [34]. This section discusses the generic case in which $D\neq 0,2(d_{S}-1)$, but it is still an integer. Within $E_{D}\left(\rho \right)$, our ensemble $\mu_{ME}^{D}$ has support on a D-dimensional submanifold of the full $2(d_{S}-1)$-dimensional quantum state space, where it has a density. Reference [34] discusses, in detail, the case in which D=1 and $d_{S}=2$. Here, we generalize the procedure to arbitrary D and $d_{S}$.

If the support of $\mu_{ME}^{D}$ is contained in a submanifold of dimension $D<2(d_{S}-1)$, which we call $S_{D}$, we can project the Fubini–Study metric $g^{FS}$ down to $P(H_{S})$ to get $g_{S}^{FS}$. Let us call $X^{j}:\xi^{a}\in S_{D}\to X^{j}\left(\xi^{a}\right)\in P\left(H_{S}\right)$ the functions which embed $S_{D}$ into the full quantum state space $P(H_{S})$. Then, the metric induced on $S_{D}$ is $g_{ab}^{S}=\sum_{j,k}\partial_{a}X^{j}\partial_{b}X^{k}g_{jk}^{FS}$, where $\partial_{a}:=\partial /\partial \xi^{a}$. Note that here we are using the “real index” notation even for coordinates $X^{j}$ on $P(H_{S})$. While $P(H_{S})$ is a complex manifold, admitting complex homogeneous coordinates, we can always use real coordinates on it. Then, $g^{S}$ induces a volume form $d\omega_{S}^{\xi }=\omega_{S}\left(d\xi \right)=\sqrt{\det g^{S}}d\xi$, where dξ is the Lebesgue measure on the $R^{D}$ which coordinatizes $S_{D}$. Then, $\mu_{ME}^{D}$ can be written as:$$\begin{array}{c} \mu_{ME}^{D}\left[A\in S_{D}\right]=\int_{A}\;\;\;d\omega_{S}^{\xi }f\left(\xi \right)\;. \end{array}$$

Eventually, this leads to:$$\begin{array}{c} h_{D}=-\int_{S_{D}}\;\;\;d\omega_{S}^{\xi }\;f\left(\xi \right)\log f\left(\xi \right)\;. \end{array}$$

This allows rewriting the constraints explicitly in a form that involves only probability densities on $S_{D}$:
$$\begin{array}{cc} C_{1} & =\mu_{ME}^{D}\left[S_{D}\right]-1=\int_{S_{D}}\;\;\;\;d\omega_{S}^{\xi }f\left(\xi \right)-1\;,\\ C_{jk} & =E_{\mu_{ME}^{D}}\left[{\left(Z^{j}\right)}^{*}Z^{k}\right]-\rho_{jk}\\ \; & =\int_{S_{D}}\;\;\;\;d\omega_{S}^{\xi }f\left(\xi \right){\left(Z^{j}\right)}^{*}\left(\xi \right)Z^{k}\left(\xi \right)-\rho_{jk}\;, \end{array}$$
where $Z\left(\xi \right):\xi \in S_{D}\to Z\left(\xi \right)\in P\left(H_{S}\right)$ are the homogeneous coordinate representation of the embedding functions $X^{a}$ of $S_{D}$ onto $P(H_{S})$.

The solution of the optimization problem leads to the Gaussian form, in homogeneous coordinates, with support on $S_{D}$:
$$\begin{array}{cc} \; & q_{ME}^{D}\left(\xi \right)=\frac{1}{Q_{D}}e^{-\sum_{j,k=1}^{d_{S}}\gamma_{jk}{\left(Z^{j}\right)}^{*}\left(\xi \right)Z^{k}\left(\xi \right)}\\ \; & Q_{D}=\int_{S_{D}}\;\;\;\;d\omega_{S}^{\xi }\;e^{-\sum_{j,k=1}^{d_{S}}\gamma_{jk}{\left(Z^{j}\right)}^{*}\left(\xi \right)Z^{k}\left(\xi \right)}\;. \end{array}$$

Again, we can move from a homogeneous representation to a symplectic one $Z\left(\xi \right)↔\left(\vec{p}\left(\xi \right),\vec{\phi }\left(\xi \right)\right)$ in which the reference basis is the eigenbasis of ρ. This gives $\rho_{jk}=\lambda_{j}\delta_{jk}$. This, in turn, means we only need the diagonal Lagrange’s multipliers $\gamma_{jj}$. As for the previous case, we move to a single label notation $\gamma_{jj}\to \gamma_{j}$:
$$\begin{array}{cc} \; & q_{ME}^{D}\left(\xi \right)=\frac{1}{Q_{D}\left(\vec{\tau }\right)}e^{\vec{\tau }\cdot \vec{p}\left(\xi \right)}\;,\\ \; & Q_{D}\left(\vec{\tau }\right)=\int_{S_{D}}\;\;\;\;\;d\omega_{S}^{\xi }\;e^{\vec{\tau }\cdot \vec{p}\left(\xi \right)}\\ \; & \vec{\tau }\left(\gamma_{d_{S}}-\gamma_{1},\gamma_{d_{S}}-\gamma_{2},\ldots ,\gamma_{d_{S}}-\gamma_{d_{S}-1}\right) \end{array}$$
with an analytical expression for the entropy:$$\begin{array}{c} h_{D}\left(\vec{\tau }\right)=\log Q_{D}\left(\vec{\tau }\right)-\vec{\tau }\cdot \vec{\lambda }\;. \end{array}$$

While this solution appears to have much in common with the $D=2(d_{S}-1)$ case, there are profound differences. Indeed, the functions $\vec{p}\left(\xi \right)$ can be highly degenerate, since we are embedding a low-dimensional manifold, $S_{D}$, into a higher one, $P(H_{S})$. Indeed, the coordinates ξ emerge from coordinatizing a submanifold of dimension D within one of dimension $2(d_{S}-1)$. This means that for $S_{D}$ there are $2(d_{S}-1)-D$ independent equations of the type ${\left\{K_{n}\left(Z\right)=0\right\}}_{n=1}^{2(d_{S}-1)-D}$. In general, we expect them to be highly nonlinear functions of their arguments. While choosing an appropriate coordinate system allows simplifying, this choice has to be made on a case-by-case basis. In specific cases, discussed in the next section, several exact solutions can be found analytically.

### 4.4. Noninteger D: Fractal Ensembles

As Ref. [34] showed, even measuring the environment in a local basis can lead to GQSs with noninteger D. For example, if we explicitly break the translational invariance of the spin-1/2 Heisenberg model in 1D by changing the local magnetic field of one spin, the GQS of one of its spin-1/2 is described by a fractal resembling Cantor’s set in the thermodynamic limit of an infinite environment. Its quantum information dimension and geometric entropy have been estimated numerically to be D≈0.83±0.02 and $h_{0.83}$ grows linearly with $N_{E}$, the size of the environment: $h_{0.83}\propto 0.66N_{E}$. Their existence gives physical meaning to the question of finding the maximum geometric entropy ensemble with noninteger D.

Providing concrete solutions to this problem is quite complex, as it requires having a generic parametrization for an ensemble with an arbitrary fractional D. As far as we know, this is currently not possible. While we do know that certain ensembles have a noninteger D, there is no guarantee that fixing the value of the information dimension, e.g., D=N/M with N,M∈N relative primes, turns into an explicit way of parametrizing the ensemble. We leave this problem open for future work.

## 5. How Does $\mu_{\mathrm{ME}}$ Emerge?

While the previous section gave the technical details regarding ensembles resulting from the proposed maximum geometric quantum entropy principle, the following identifies the mechanisms for their emergence in a number of cases of physical interest.

### 5.1. Emergence of $\mu_{ME}^{0}$

As partly discussed in the previous section, $\mu_{ME}^{0}$ can emerge naturally as a conditional ensemble, when our system of interest interacts with a finite-dimensional environment (dimension *N*). If the environment is probed with projective measurements in a basis that is unbiased with respect to the Schmidt-partner basis ${\left\{|SP_{\alpha }\rangle \right\}}_{\alpha =1}^{N}$, we reach the absolute maximum of the geometric entropy, logN. The resulting GQS is $\mu_{ME}^{0}=\delta_{\vec{\lambda }}^{\vec{p}}\frac{1}{N}\sum_{\alpha =1}^{N}\delta_{{\vec{\theta }}_{\alpha }}^{\vec{\phi }}$, with members of the ensemble being $|\chi_{\alpha }\rangle \;=\sum_{j=1}^{d_{S}}\sqrt{\lambda_{j}}e^{i\theta_{\alpha j}}|\lambda_{j}\rangle$ and $p_{\alpha }=1/N$.

As argued in Ref. [41], the notion of unbiasedness is typical. Physically, this is interpreted as follows. Imagine someone gives up |ψ(ρ)〉, a purification of ρ, without telling us anything about the way the purification is performed. This means we know nothing about the way ρ has been encoded into |ψ(ρ)〉. Equivalently, we do not know what the ${\left\{|SP_{j}\rangle \right\}}_{j=1}^{d_{S}}$ are. If we now choose a basis of the environment to study the conditional ensemble, ${\left\{|v_{\alpha }\rangle \right\}}_{\alpha =1}^{N}$, this will have very little information about the ${\left\{|SP_{j}\rangle \right\}}_{j=1}^{d_{S}}$—there is a very high chance that we will end up very close to the unbiasedness condition.

The mathematically rigorous version of “very high chance” and “very close” is given in Ref. [41] and it is not relevant here. The only thing we need is that this behavior is usually exponential in the size of the environment $∼2^{N}$. Somewhat more accurately, the fraction of bases which are ${\left|\langle v_{\alpha }|SP_{j}\rangle \right|}^{2}\approx 1/N$ are $∼1-2^{-N}$. Therefore, statistically speaking, it is extremely likely that, in absence of meaningful information about what the ${\left\{|SP_{j}\rangle \right\}}_{j=1}^{d_{S}}$ are, the conditional ensemble we will see is $\mu_{ME}^{0}$.

### 5.2. Emergence of $\mu_{ME}^{2(d_{S}-1)}$

For $\mu_{ME}^{2(d_{S}-1)}$ to emerge as a conditional ensemble, our $d_{S}$-dimensional quantum system must interact with an environment that is being probed with measurements whose outcomes are parametrized by $2(d_{S}-1)$ continuous variables, each with the cardinality of the reals. This is because we have to guarantee that $D=2(d_{S}-1)$. Therefore, conditioning on projective measurements on a finite environment is insufficient. One possibility is to have a finite environment that we measure on an overcomplete basis, like coherent states. A second possibility is to have a genuinely infinite-dimensional environment, on which we perform projective measurements. For example, we could have $2(d_{S}-1)/3$ quantum particles in 3D that we measure on the position basis $\left\{{⊗}_{n=1}^{2(d_{S}-1)/3}|x_{n},y_{n},z_{n}\rangle \right\}$. All the needed details were given in Ref. [6], where we studied the properties of a GQS emerging from a finite-dimensional quantum system interacting with one with continuous variables.

We stress here that this is only a necessary condition, not a sufficient one. Indeed, we can have an infinite environment that is probed with projective measurements on variables with the right properties but still obtain an ensemble that is not $\mu_{ME}^{2(d_{S}-1)}$. An interesting example of this is given by the continuous generalization of the notion of unbiased basis. We illustrate this in a simple example of a purification obtained with a set of $2(d_{S}-1)$ real continuous variables, realized by $2(d_{S}-1)$ non-interacting particles in a 1D box [0,L].

In this, the notion of an unbiased basis is satisfied by position and momentum eigenstates: $\langle \vec{x}|\vec{k}\rangle =\frac{e^{i\vec{k}\cdot \vec{x}}}{\sqrt{V}}$. Thus, if our Schmidt partners are momentum eigenstates ${\left\{|SP_{j}\rangle \;=\;|{\vec{k}}_{j}\rangle \right\}}_{j=1}^{d_{S}}$, and we measure the environment in the position basis, we do not obtain a GQS with the required $D=2(d_{S}-1)$. Indeed, while we do obtain $q\left(\vec{x}\right)=\langle \psi \left(\rho \right)|I_{S}⊗|\vec{x}\rangle \langle \vec{x}||\psi \left(\rho \right)\rangle \;=\frac{1}{V}$, the members of the ensemble $|\chi \left(\vec{x}\right)\rangle \;=\sum_{j=1}^{d_{S}}\sqrt{\lambda_{j}}e^{i{\vec{k}}_{j}\cdot \vec{x}}|\lambda_{j}\rangle$ are not distributed in the appropriate way.

This leads to the ensemble $\delta_{\vec{\lambda }}^{\vec{p}}\;\frac{1}{{\left(2\pi \right)}^{d_{S}-1}}$, which has the wrong information dimension: $D=d_{S}-1$, not $D=2(d_{S}-1)$. This clarifies why, in order to have $D=2(d_{S}-1)$, using an environmental basis that is unbiased with respect to the Schmidt partners is not enough. Specifically, the probabilities $p_{j}\left(\vec{x}\right)={\left|\langle \lambda_{j}|\chi \left(\vec{x}\right)\rangle \right|}^{2}=\lambda_{j}$ do not depend on $\vec{x}$. They do not obtain redistributed by the unbiasedness condition and are always equal to the eigenvalues of ρ.

If we measure on a different basis $\left\{|\vec{l}\rangle \;:=\int_{V}d\vec{x}\;u_{\vec{l}}^{*}\left(\vec{x}\right)\right\}$, we obtain a different GQS since $\langle \vec{l}|SP_{j}\rangle =\int_{V}d\vec{x}u_{\vec{l}}\left(\vec{x}\right)e^{i{\vec{k}}_{j}\cdot \vec{x}}=F_{\vec{l}}\left({\vec{k}}_{j}\right)$ is essentially the Fourier transform of $u_{\vec{l}}$:(4)$$\begin{array}{cc} \; & q\left(\vec{l}\right)=\sum_{j=1}^{d_{S}}\lambda_{j}{\left|F_{\vec{l}}\left({\vec{k}}_{j}\right)\right|}^{2} \end{array}$$(5)$$\begin{array}{cc} \; & |\chi \left(\vec{l}\right)\rangle \;=\sum_{j=1}^{d_{S}}\sqrt{\lambda_{j}}\frac{F_{\vec{l}}\left({\vec{k}}_{j}\right)}{\sqrt{q(\vec{l})}}|\lambda_{j}\rangle \;. \end{array}$$

Equation (Equation 5) gives the functions ${\left\{p_{j}\left(\vec{l}\right),\phi_{j}\left(\vec{l}\right)\right\}}_{j=1}^{d_{S}-1}$: (6)$$\begin{array}{cc} \; & p_{j}\left(\vec{l}\right)=\frac{\lambda_{j}{\left|F_{\vec{l}}\left({\vec{k}}_{j}\right)\right|}^{2}}{\sum_{n=1}^{d_{S}}\lambda_{n}{\left|F_{\vec{l}}\left({\vec{k}}_{n}\right)\right|}^{2}}\;, \end{array}$$(7)$$\begin{array}{cc} \; & \phi_{j}\left(\vec{l}\right)=\mathrm{Arg}\left(F_{\vec{l}}\left({\vec{k}}_{j}\right)\right)\;. \end{array}$$

This, together with the density $q(\vec{l})$ specifies the ensemble via $\mu =\int d\vec{l}q\left(\vec{l}\right)\delta_{\vec{p}\left(\vec{l}\right)}^{\vec{p}}\delta_{\vec{\phi }\left(\vec{l}\right)}^{\vec{\phi }}$.

Finding the exact conditions that lead to $\mu =\mu_{ME}^{2(d_{S}-1)}$ involves solving a complex inverse problem. However, what we have accomplished so far allows us to understand the real mechanism behind its emergence. First, the $\phi_{j}\left(\vec{l}\right)$ must be uniformly distributed: they must be random phases. Second, the distribution of $\vec{p}$ must be of exponential form. The first condition can always be ensured by choosing some $\langle \vec{l}|\vec{x}\rangle =u_{\vec{l}}\left(\vec{x}\right)$ and then multiplying it by pseudo-random phases, generated in a way that is completely independent on $\vec{p}$. This can always be achieved without breaking the unitarity of $\langle \vec{l}|\vec{x}\rangle$ via $u_{\vec{l}}\left(\vec{x}\right)\to u_{\vec{l}}\left(\vec{x}\right)e^{i\theta_{\vec{l}}}$. This guarantees that the marginal distribution over the phases is uniform and that the density $q(\vec{p},\vec{\phi })$ becomes a product of its marginals, since the distribution of the $\vec{\phi }$ has been built to be independent of everything else: $q\left(\vec{p},\vec{\phi }\right)=f\left(\vec{p}\right)\;\cdot \;\mathrm{unif}\left(\vec{\phi }\right)$. Then, in order for $q(\vec{p},\vec{\phi })$ to be the maximum entropy one we need $f\left(\vec{p}\right)=\frac{1}{Q(\vec{\tau })}e^{\vec{\tau }\cdot \vec{p}}$.

Given a nondegenerate $\vec{p}\left(\vec{l}\right)$, this can be ensured by a specific form of $q(\vec{l})$ since $f\left(\vec{p}\right)=\int d\vec{l}q\left(\vec{l}\right)\delta_{\vec{p}\left(\vec{l}\right)}^{\vec{p}}$:$$\begin{array}{c} q\left(\vec{l}\right)=\left(\det J\right)\;\left(\vec{l}\right)\;\frac{e^{\vec{\tau }\cdot \vec{p}\left(\vec{l}\right)}}{Q(\tau )}\;\Rightarrow \;f\left(\vec{p}\right)=\frac{e^{\vec{\tau }\cdot \vec{p}}}{Q(\tau )}\;, \end{array}$$
where *J* is the Jacobian matrix of the coordinate change $\vec{l}\to \vec{p}\left(\vec{l}\right)$. Checking that this form leads to the right distribution is simply a matter of coordinate changes. Alternatively, it can be seen by repeated use of the Laplace transform on the simplex, together with the result $\frac{1}{Q(\vec{\tau })}\int_{\Delta_{d_{S}-1}}\;\;\;d\vec{p}e^{-\vec{a}\cdot \vec{p}}e^{\vec{\tau }\cdot \vec{p}}=Q\left(\vec{\tau }-\vec{a}\right)/Q\left(\vec{\tau }\right)$. We now see the mechanism at play in a concrete way and how it leads to the maximum entropy GQS $\mu_{ME}^{2(d_{S}-1)}$.

First, let us take the label $\vec{l}=\left(l_{1},\ldots ,l_{2(d_{S}-1)}\right)$ and split it in two $\vec{l}=\left(\vec{a},\vec{b}\right)$ with $\vec{a}=\left(a_{1},\ldots ,a_{d_{S}-1}\right)$ and $\vec{b}=\left(b_{1},\ldots ,b_{d_{S}-1}\right)$. Then, $d\vec{l}=d\vec{a}d\vec{b}$. At this stage, the choice of $\vec{l}$, the splitting, and $|SP_{j}\rangle$ are arbitrary. Then, we make the choice that $\langle \vec{a},\vec{b}|SP_{j}\rangle$ $=\sqrt{A_{j}\left(\vec{a}\right)}e^{iB_{j}\left(\vec{b}\right)}$. The only property we need to check is that ${\left\{|\vec{a},\vec{b}\rangle \right\}}_{\vec{a},\vec{b}}$ can be a complete set:$$\begin{array}{c} \int d\vec{a}\sqrt{A_{j}\left(\vec{a}\right)A_{k}\left(\vec{a}\right)}\int d\vec{b}e^{i(B_{j}\left(\vec{b}\right)-B_{k}\left(\vec{b}\right))}=\delta_{jk}\;. \end{array}$$

We can choose $B_{j}\left(\vec{b}\right)$ such that $\int d\vec{b}e^{i(B_{j}\left(\vec{b}\right)-B_{k}\left(\vec{b}\right))}=M\delta_{jk}$, for example, by choosing $B_{j}\left(\vec{b}\right)$ to be linear functions. Then, choosing $A_{j}\left(\vec{a}\right)$ such that $\int d\vec{a}A_{j}\left(\vec{a}\right)=\frac{1}{M}$ guarantees completeness. With this choice, we obtain:$$\begin{array}{cccc} \; & q\left(\vec{a},\vec{b}\right)=\sum_{j=1}^{d_{S}}\lambda_{j}A_{j}\left(\vec{a}\right)\;,\\ \; & p_{j}\left(\vec{a},\vec{b}\right)=\frac{\lambda_{j}A_{j}\left(\vec{a}\right)}{\sum_{n=1}^{d_{S}-1}\lambda_{n}A_{n}\left(\vec{a}\right)} & \; & \to p_{j}\left(\vec{a}\right)\;,\\ \; & \phi_{j}\left(\vec{a},\vec{b}\right)=B_{j}\left(\vec{b}\right) & \; & \to \phi_{j}\left(\vec{b}\right)\;. \end{array}$$

The probability density $q(\vec{a},\vec{b})$ can be written as a product of two probability densities: $q\left(\vec{a},\vec{b}\right)=f\left(\vec{a}\right)\frac{1}{M}$. Here, 1/M is the uniform density for $\vec{b}$ and $f\left(\vec{a}\right)=\sum_{j=1}^{d_{S}}\lambda_{j}MA_{j}\left(\vec{a}\right)$ is a probability density for $\vec{a}$. Then, the GQS becomes a product of two densities: one over the probability simplex (for $\vec{p}$) and another one over the phases (for $\vec{\phi }$):(8)$$\begin{array}{cc} \mu & =\;\;\int \;\;d\vec{a}\;\;\int \;\;d\vec{b}\;q\left(\vec{a},\vec{b}\right)\;\delta_{\vec{p}\left(\vec{a},\vec{b}\right)}^{\vec{p}}\delta_{\vec{\phi }\left(\vec{a},\vec{b}\right)}^{\vec{\phi }}\;,\\ \; & =\int d\vec{a}f_{a}\left(\vec{a}\right)\delta_{\vec{p}\left(\vec{a}\right)}^{\vec{p}}\;\cdot \;\frac{1}{M}\int d\vec{b}\delta_{\vec{\phi }\left(\vec{b}\right)}^{\vec{\phi }}\;,\\ \; & =f_{p}\left(\vec{p}\right)f_{\phi }\left(\vec{\phi }\right)\;. \end{array}$$

These are the formulas for two changes of variable in integrals: $\vec{a}\to \vec{p}\left(\vec{a}\right)$ and $\vec{b}\to \vec{\phi }\left(\vec{b}\right)$. Since these are invertible, we can confirm what we understood before. $f_{\phi }\left(\vec{\phi }\right)=\mathrm{unif}\left[\vec{\phi }\right]$ when the phases $\vec{\phi }\left(\vec{b}\right)$ are uniformly distributed. Moreover, when $f_{p}\left(\vec{p}\right)=\frac{e^{\tau \cdot \vec{p}}}{Q(\vec{\tau })}$ and $\vec{p}\left(\vec{a}\right)$ are exponentially distributed: $f_{a}\left(\vec{a}\right)=\left(\det J_{ap}\right)\left(\vec{a}\right)\frac{e^{\vec{\tau }\cdot \vec{p}\left(\vec{a}\right)}}{Q(\vec{\tau })}$, with $J_{ap}$ being the Jacobian matrix of the change of variables $\vec{a}\to \vec{p}\left(\vec{a}\right)$.

### 5.3. Stationary Distribution of Some Dynamic

A second mechanism, which can lead to the emergence of an ensemble with $D=2(d_{S}-1)$, is time averaging. Indeed, if we are in a nonequilibrium dynamical situation in which the system and its environment jointly evolve with a dynamical (possibly unitary) law, its conditional ensembles μ(t) depend on time.

To study stationary behavior from dynamics, one looks at time-averaged $\bar{\mu (t)}$ $=\lim_{T\to \infty }\frac{1}{T}\int_{0}^{T}\mu \left(t\right)$ ensembles that, in this case, have a certain stationary density matrix $\rho_{ss}=\bar{\rho (t)}$. Unless something peculiar happens, we expect the ensemble to cover large regions of the full state space, leading to a stationary GQS with $D=2(d_{S}-1)$ and a given density matrix $\rho_{ss}$.

Intuitively, we expect dynamics that are chaotic in quantum state space to lead to ensembles described by $\mu_{ME}^{2(d_{S}-1)}$. This is because the ensemble that emerges must be compatible with a density matrix $\rho_{ss}$ while still exhibiting a nontrivial dynamics due to the action of the environment. We now give a simple example of how this happens. Borrowing from Geometric Quantum Thermodynamics, see Ref. [7], where we studied a qubit with a Caldeira–Leggett-like environment, the resulting evolution for the qubit can be described using the Stochastic Schrödinger equation, which, as shown in Ref. [7], leads to a maximum entropy ensemble (see Equation (3)) of the required type.

### 5.4. Emergence of $\mu_{ME}^{d_{S}-1}$

Among all possible values of D, a third one which is particularly relevant is $D=d_{S}-1$, which is half the maximum value. The reason why this is important comes from the symplectic nature of the quantum state space and, ultimately, from dynamics. One physical situation in which $\mu_{ME}^{d_{S}-1}$ emerges naturally is the study of the dynamics of pure, isolated quantum systems. The phenomenology we discuss here is known, being intimately related to thermalization and equilibration studies. We discuss it here only in connection with the maximum geometric entropy principle introduced in Section 4.

Imagine an isolated quantum system in a pure state $|\psi_{0}\rangle$ evolving unitarily with a dynamics generated by some time-independent Hamiltonian $H=\sum_{n=1}^{D}E_{n}|E_{n}\rangle \langle E_{n}|$. Assuming lack of degeneracies in the energy spectrum, the dynamics is given by $|\psi_{t}\rangle =\sum_{n=1}^{D}\sqrt{p_{n}^{0}}e^{i(\phi_{n}^{0}-E_{n}t)}$, where $\sqrt{p_{n}^{0}}e^{i\phi_{n}^{0}}\langle E_{n}|\psi_{0}\rangle$ and we have used symplectic coordinates in the energy eigenbasis. Since $\vec{p}\in \Delta_{D-1}$ are conserved quantities $p_{n}^{t}=p_{n}^{0}$ and $\vec{\phi }\in T_{D-1}$ evolve independently and linearly $\phi_{n}^{t}=\phi_{n}^{0}-E_{n}t$ on a high-dimensional torus, we know that a sufficient condition for the emergence of ergodicity on $T_{D-1}$ is the so-called non-resonance condition: energy gaps have to be non-degenerate: namely $E_{n}-E_{k}=E_{a}-E_{b}$ if and only if n=k and a=b or n=a and k=b.

This condition is usually true for interacting many-body quantum systems. If that is the case, then the evolution of the phases is ergodic on $T_{D-1}$. This was first proven by von Neumann [42] in 1929. Calling $\left(\vec{p}\left(t\right),\vec{\phi }\left(t\right)\right)$ the instantaneous state and with $\delta_{\vec{p}\left(t\right)}^{\vec{p}}\delta_{\vec{\phi }\left(t\right)}^{\vec{\phi }}$ the corresponding Dirac measure on the quantum state space, we have:$$\begin{array}{c} \lim_{T\to \infty }\frac{1}{T}\int_{0}^{T}dt\delta_{\vec{p}\left(t\right)}^{\vec{p}}\delta_{\vec{\phi }\left(t\right)}^{\vec{\phi }}=\delta_{\vec{p}\left(0\right)}^{\vec{p}}\;\cdot \;{\mathrm{unif}}^{\vec{\phi }}\left(T_{D-1}\right)\;, \end{array}$$
where ${\mathrm{unif}}^{\vec{\phi }}\left(T_{D-1}\right)$ is the uniform measure on $T_{D-1}$ in which all $\phi_{n}$ are uniformly and independently distributed on the circle. It is not too hard to see that this is the maximum geometric entropy ensemble with $D=d_{S}-1$, compatible with the fact that the occupations of the energy eigenstates are all conserved quantities: $p_{n}^{t}=p_{n}^{0}$.

Indeed, these $d_{S}-1$ constraints provide $d_{S}-1$ independent equations, thus reducing the state-space dimension that the system explores to the high-dimensional torus $T_{D-1}$. On this, however, the dynamics is ergodic and the resulting stationary measure is the uniform one. By definition, this is the measure with the highest possible value of geometric entropy since its density is uniform and equal to $q_{ME}^{\phi }\left(\vec{\phi }\right)=\frac{1}{\mathrm{Vol}[T_{D-1}]}$, where $\mathrm{Vol}[T_{D-1}]$ $=\int_{T_{D-1}}\;\;\;d\omega_{FS}^{\phi }={\left(2\pi \right)}^{D-1}$ is the volume of $T_{D-1}$, computed with the Fubini–Study volume element projected on $T_{D-1}$, that is $\prod_{k=1}^{D-1}d\phi_{k}$:$$\begin{array}{cc} h_{d_{S}-1} & =-\int_{T_{D-1}}\;\;\;d\omega_{FS}^{\phi }\frac{1}{\mathrm{Vol}[T_{D-1}]}\log\frac{1}{\mathrm{Vol}[T_{D-1}]}\\ \; & =\log\mathrm{Vol}[T_{D-1}]\;. \end{array}$$

### 5.5. Comment on the Generic $\mu_{ME}^{D}$

To have a GQS with generic information dimension D result from a conditional measurement on an environment, we must condition on at least D continuous variables with the cardinality of the reals. This can be achieved either via measurements on an overcomplete basis, such as coherent states, or via projective measurements on an infinite dimensional environment with at least D real coordinates. This condition is necessary, but not sufficient, to guarantee the emergence of the corresponding maximum entropy ensemble $\mu_{ME}^{D}$. While we have seen that the notion of an unbiased basis is relevant when D=0, we also saw how this falls short in the generic D>0 case. Understanding this general condition is a nontrivial task that requires a much deeper understanding of how systems encode quantum information in their environment and how this is extracted by means of quantum measurements. Further work in this direction is ongoing and will be reported elsewhere.

For a GQS with arbitrary dimension D to emerge as a stationary distribution on a quantum state space with dimension $2(d_{S}-1)$, it is likely that we need $2(d_{S}-1)-D$ independent equations constraining the dynamics, that is, if D is an integer. Indeed, due to the continuity of time and the smoothness of the time evolution in quantum state space, we expect D∈N in the vast majority of cases. If these equations constraining the motion on the quantum state space are linear, then we know that having $2(d_{S}-1)-D$ independent equations is both necessary and sufficient to have D as quantum information dimension. This, however, says virtually nothing about the maximization of the relevant geometric entropy $h_{D}$. Moreover, constraints on an open quantum system can take very generic forms and the relevant equations will not always be linear.

An explicit example where such ensemble can be found constructively is given by the case of an isolated quantum system. The conditions for the emergence of a maximum entropy $\mu_{ME}^{d_{S}-1}$ are then known, being equivalent to the conditions for the ergodicity of periodic dynamics on a high-dimensional torus, which are known.

## 6. Conclusions

While a density matrix encodes all the statistics available from performing measurements on a system, they do not give information about how the statistics were created. And infinite possibilities are available. A natural way to select a unique ensemble behind a given density matrix is to approach the problem from the perspective of information theory. In this case, the issue becomes a standard inference problem, one to which we can apply the maximum entropy principle. To properly formulate the problem in this way requires a proper way to compute the entropy of an ensemble. While this is trivial for ensembles with a finite number of elements, it is not for continuous ensembles. The correct answer, the notion of Geometric Quantum Entropy $h_{D}$, was given in Ref. [34]. This, however, depends strongly on another quantity that characterizes the ensemble: the quantum information dimension D. Consequently, we formulated the maximum geometric entropy principle at fixed quantum information dimension. This is a one-parameter class of maximum entropy principles, labeled by D, that can be used to explore various ways to have ensembles give rise to a specific density matrix.

As often happens with inference principles, the generic optimization problem can be hard to solve. However, here we solved a number of cases where the ensemble can be found analytically. We also explored the physical mechanism responsible for the emergence of $\mu_{ME}^{D}$. Two different classes of situations were considered: (i) a conditional ensemble, resulting from measuring the environment of our system of interest, and (ii) stationary distributions, in which the statistics arise from aggregating data over time. We have also identified and discussed various instances where both mechanisms lead to a maximum entropy ensemble.

## Data Availability

The data that support the findings of this study are available from the corresponding author upon reasonable request.

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
