# Peer review of "Maximum Geometric Quantum Entropy"

_entropy, 2024, doi:10.3390/e26030225_

Round 1
Reviewer 1 Report
Comments and Suggestions for Authors
This paper extends the authors' previous work, published in Physical Review, to develop a formulation of statistical thermodynamics from geometric quantum mechanics. The present paper focuses on the identification of the geometric quantum state by maximizing the quantum entropy. As the authors point out, in many respects this work parallels that of Brody and Hughston (ref 27) but enforces a more rigorous and general definition of the entropy to accomplish these ends.
I find no fault with the goals or the reasoning. I confess that I am not qualified to evaluate the section on fractal ensembles. The paper is well-constructed with attention to instructive examples (e.g. the example of qubit states to illustrate the need for a more general formulation). The integration of results from prior literature into the present work is thoughtful and informative, avoiding needless repetition.
My only criticism is quite minor. There is some odd formatting in a few of the references. It appears that several give specific page numbers within the article (which is unusual for citations of journal articles), and a few fail to give article numbers (refs 6, 7, 25, 26, 34, maybe others). There are also spelling mistakes in the references ("quatum", "exhcange") whereas the rest of the article is very carefully proofread.
Reviewer 2 Report
Comments and Suggestions for Authors
The authors consider the problem of identifying the different ensembles on the set of quantum states which produce a given density matrix. For this purpose, they consider a version of the Maximum entropy principle adapted to the geometric formulation of Quantum Mechanics. From a technical point of view, their approach is based on the Hughston-Josza-Wootters (HJW) theorem and the definition of geometric quantum states as probability measures on the set quantum states (the projective space corresponding to a given Hilbert space). The paper is interesting and it is well written. Nonetheless, there are certain points which I would like to understand better before recommending the acceptance. Let me summarize them briefly:
A) When using probability measures on the projective space, one must be careful with the notion of mutual exclusivity in quantum systems. When representing a state with a density matrix, rho captures that notion in the orthogonality of the different eigenspaces, but when using geometric measures the problem is more difficult. The corresponding entropy function must take this issue into consideration and I do not understand how the definition of the geometric entropy function 1b does that. Could the authors include a discussion to explain it?
B) Furthermore, I do not understand the physical meaning of the geometric entropy function and/or the maximum entropy principle. Should we expect it to produce a well-defined thermodynamical entropy function and ensemble for the system in any case? Can the authors explain it? My main concern in this sense refers to the "Full support" case when D=2(d_S+1), in page 6. If I understand correctly, the authors claim that they recover the result in [27] where the resulting probability measure is claimed to represent a notion of canonical ensemble. Nonetheless, such claim is rejected in Reference [26], Section 6.1.5. Also in https://doi.org/10.1103/PhysRevE.91.022137, it is argued that this distribution exhibits properties which make them incompatible with equilibrium thermodynamics. Do these effects appear in the other solutions presented? Does this represent a limitation for the physical systems this approach can be applied to?
Reviewer 3 Report
Comments and Suggestions for Authors
The authors discuss the ambiguity in the ensemble description of any given density matrix.
The manuscript addresses one of the most fundamental properties of quantum mechanics, proposing a thought-provoking view point on it.
Among infinite compatible ensembles, the authors pick those that maximize the so-called geometric quantum entropy, which the same authors have previously introduced in a previous work.
I find this to be a valuable contribution and therefore recommend the manuscript for publication.
Here are two questions for the authors:
1. One of the most important applications of quantum information theory is quantum cryptography.
In quantum cryptography, the environment (which holds the purifying system) is considered to be ad adversary.
Therefore, to account for the worst-case scenario, the entropy of the ensemble decomposition is, in quantum cryptography, always minimized, and not maximized as in this work.
What do the authors think about this?
2. I wonder if this approach can be put in relation with the work of Tim Palmer
(see for example https://arxiv.org/abs/2204.05763 and references therein)
Round 2
Reviewer 2 Report
Comments and Suggestions for Authors
The authors have answered satisfactorily all my comments. I recommend the acceptance of the paper.